

# Long-term macrobioerosion in the Mediterranean Sea assessed by micro-computed tomography

C. Färber[1], J. Titschack[1,2], C.H.L. Schönberg[3, 4], K. Ehrig[5], K. Boos[2], D. Baum[6], B. Illerhaus[5],
U. Asgaard[7], R.G. Bromley[7], A. Freiwald[1], M. Wisshak[1]

[1]Senckenberg am Meer, Abteilung Meeresforschung, Südstrand 40, 26382 Wilhelmshaven, Germany
[2]MARUM, Center for Marine Environmental Sciences, Leobener Straße, 28359 Bremen, Germany
[3]Oceans Institute, The University of Western Australia, Crawley, WA 6009, Australia
[4]Western Australian Museum, Welshpool, WA 6106, Australia
[5]Bundesanstalt für Materialforschung und –prüfung (BAM), Unter den Eichen 87, 12205 Berlin, Germany
[6]ZIB, Zuse Institute Berlin, Takustraße 7, 14195 Berlin, Germany
[7]Rønnevej 97, DK-3720 Aakirkeby, Denmark

*Correspondence to*: Claudia Färber (claudia.faerber@senckenberg.de)

**Abstract.** Biological erosion is a key process for the recycling of carbonate and the formation of calcareous sediments in the oceans. Experimental studies showed that bioerosion is subject to distinct temporal variability, but previous long-term studies were restricted to tropical waters. Here, we present results from a 14 year bioerosion experiment that was carried out along the rocky limestone coast of the island of Rhodes, Greece, in the Eastern Mediterranean Sea, in order to monitor the pace at which bioerosion affects carbonate substrate and the sequence of colonisation by bioeroding organisms. Internal macrobioerosion was visualised and quantified by micro-computed tomography and computer-algorithm based segmentation procedures. Analysis of internal macrobioerosion traces revealed a dominance of bioeroding sponges producing eight types of characteristic *Entobia* cavity networks, which were matched to five different clionaid sponges by spicule identification in extracted tissue. The morphology of the entobians strongly varied depending on the species of the producing sponge, its ontogenetic stage, available space, and competition by other bioeroders. An early community developed during the first 5 years of exposure with initially very low macrobioerosion rates and was followed by an intermediate stage when sponges formed large and more diverse entobians and bioerosion rates increased. After 14 years, 30 % of the block volumes were occupied by boring sponges, yielding maximum bioerosion rates of 900 g m$^{-2}$ yr$^{-1}$. A high spatial variability in macrobioerosion prohibited clear conclusions about the onset of macrobioerosion equilibrium conditions. This highlights the necessity of even longer experimental exposures and higher replication at various factor levels in order to better understand and quantify temporal patterns of macrobioerosion in marine carbonate environments.

## 1 Introduction

Bioerosion, the erosion of hard substrate by living organisms (Neumann, 1966), generally comprises (i) internal microbioerosion by boring cyanobacteria, algae, and fungi, (ii) internal macrobioerosion by boring sponges, worms, and bivalves, and (iii) external bioerosion by grazing gastropods, echinoids, and fish (e.g., Glynn, 1997; Tribollet et al., 2011).



Experimental studies showed that the succession of bioerosion agents is subject to distinct temporal variability: Under favourable conditions, microborers can reach stable communities within 1 year of exposure (Grange et al., 2015; Tribollet and Golubic, 2005), whereas establishment of mature communities of macrobioeroders may take several years to form mature communities (e.g., Chazottes et al., 2002; Kiene and Hutchings, 1992, 1994; Pari et al., 1998, 2002; Tribollet and

Golubic, 2005). Most bioerosion experiments were conducted over a period of only 1-2 years, giving a detailed picture on microbioerosion in different geographical settings (e.g., Kiene, 1988; Vogel et al., 1996, 2000; Wisshak, 2006; Wisshak et al., 2010, 2011). Experimental studies on the succession of macrobioeroders were previously limited to tropical coral reef systems and commonly lasted about 4-8 years (Carreiro-Silva and McClanahan, 2012; Kiene and Hutchings, 1992, 1994; Pari et al., 2002). The longest experiments have been conducted over 12 years at the Great Barrier Reef (Kiene and

Hutchings, 1992) and 13 years at Jamaica (Scott et al., 1988). To date, no experimental data on long-term bioerosion from non-tropical settings are available, but would constitute important information for evaluating global patterns of bioerosion and for modelling future impacts of bioerosion. This is particularly relevant since bioerosion is considered to increase with ongoing ocean acidification (Tribollet et al., 2009), a trend that is especially true for bioeroding sponges (e.g., Fang et al., 2013; Wisshak et al., 2012, 2013, 2014), with potentially detrimental effects on carbonate-dominated ecosystems (Kennedy

et al., 2013).

In the Mediterranean Sea, bioerosion affects sensitive ecosystems such as limestone coasts, deposits of coralline algae, and cold-water coral reefs, as well as molluscs in aquaculture, submerged man-made materials, and artefacts (see Schönberg and Wisshak, 2014 for a review). Experimental data on Mediterranean bioerosion are only available in form of short-term observations on microendoliths (Färber et al., 2015; Le Campion-Alsumard, 1979). Here, we present results from a long-

term bioerosion experiment that was carried out over 14 years along the limestone rocky shore of Rhodes (Greece) in order to analyse the succession of bioeroders in the Eastern Mediterranean Sea. A preliminary summary on macroscopic observations during the first 6 years was provided by Bromley et al. (1990). For the visualisation of internal macrobioerosion traces and quantification of macrobioerosion rates, micro-computed tomographic analysis was chosen as a non-destructive approach. Computed tomography is increasingly used to visualise bioerosion traces in three dimensions (Beuck et al., 2007,

2008; Bromley et al., 2008; Schönberg and Shields, 2008), but quantitative approaches are still scarce and comparatively new (Crook et al., 2013; DeCarlo et al., 2015; Silbiger et al., 2014). Aim of this paper is (i) to introduce a novel approach to visualise and quantify internal bioerosion using computer-algorithm based segmentation procedures, (ii) to provide an inventory of macrobioerosion traces, (iii) to identify trace making boring sponges through spicule analysis, and (iv) to assess the long-term development of bioerosion rates and ontogenetic development of sponge borings in terms of a possible onset

of macrobioerosion equilibrium conditions.



## 2 Material and Methods

### 2.1 Experimental design

Experimental blocks were deployed in the vicinity of four limestone cliffs at the east and west coast of the island of Rhodes, Greece (Fig. 1): (i) at the south and east edge of Ladiko Bay (36°19'5"N, 28°12'17"E; 36°19'10"N, 28°12'29"E), (ii) south of Kolimbia (36°14'26"N, 28°9'44"E; 36°14'21"N, 28°9'47"E), (iii) north of St. Paul's Bay near Lindos (36°5'17"N, 28°5'20"E), and (iv) in Pyrgos (36°10'10"N, 27°43'55"E). All localities were characterised by limestone rock ground or boulder fields, and were free from local pollution. Annual monitoring in October showed no indication of interference of the experiment by human activities. The blocks were of pure calcium carbonate (marble and limestone), and respective lithology was confirmed by petrographic thin sections. Between 1982 and 1989, 46 blocks were placed directly on the sea floor, in water depths between 3 and 17 m, and until 1996 each year some blocks were retrieved, all by skin diving and using floatation devices. Recovered blocks were rinsed in fresh water and soft epiliths were removed. The blocks were subsequently photographed and dried. From all retrieved blocks, for the present approach twelve were chosen from different depths (3 to 17 m) and exposure times (1 to 14 years), all with a surface area of 11 x 11 cm or more. Most of these showed no evidence for having been dislocated during the experiment, however, some blocks had slightly moved and for some the information about recovery date and/or water depth was incomplete (Table 1). This was reviewed case by case, and ultimately the latter blocks were included in the study.

### 2.2  Micro-computed tomography

Internal bioerosion in the blocks was investigated by micro-computed tomographic analysis (micro-CT). In order to yield spatial resolutions of about 70 µm, blocks were cut with a rock saw to a uniform surface area of 10 x 10 cm. The 7-year block was cut into three such replicates to obtain an impression of the spatial variability of bioerosion within a single block. In this way, a total of 14 samples (in the following called blocks) were selected for micro-CT analysis.

Micro-CT scanning was carried out at the Bundesanstalt für Materialforschung und –prüfung (BAM), Berlin, Germany, using the 225 kV system (Badde and Illerhaus, 2008). An X-ray source voltage of 210 kV, a current of 90 µA, and a pre-filter of 1 mm copper was applied. Attenuation images were taken at smallest possible resolution due to specimen size. To achieve the best signal-to-noise-ratio, 2400 projections over 360 degrees with a total measuring time of 16 h were taken. Images were reconstructed using BAM software generated from the original Feldkamp algorithm (Feldkamp et al., 1984). The resulting voxel size was 72 µm.

Post-processing of micro-CT data was conducted using the Amira software edition from the Zuse Institute Berlin, ZIBAmira version 2014.51 (Stalling et al., 2005). In an initial segmentation step all encrusting epiliths on the surface of the blocks were excluded from the dataset using the *Segmentation Editor*. Limestone substrate was distinguished from the surrounding air and organic tissue (borings were partially filled by air and organic remains of the sponges) using the marker-based





*Watershed* segmentation module. Segmentation of the borings from the space surrounding the block was carried out with the *AmbientOcclusionField* module (Baum & Titschack, subm.). Resulting micro-CT images of the blocks were cropped to uniform sizes of 90 x 90 x 18 mm with the *CropEditor* to obtain comparable volumes. The respective volumes of substrate and bioerosion per block were quantified with the *MaterialStatistics* module using the results from the latter segmentation.

To quantify the total surface area of each block, bioerosion and substrate were selected together and the surface was calculated using the *SurfaceGen* module. After removing all other surfaces except the upper surface with the *SurfaceEditor*, this surface area was quantified with the *SurfaceStatistics* module. To further evaluate the bioerosion constituents, a third segmentation step was performed based on a *DistanceMap* of the segmented bioerosion traces employing the *ContourTreeSegmentation* module to gain an automatic separation of different bioerosion traces in the blocks (threshold: 0,

persistence value: 0.05; see Titschack et al., 2015). Subsequently, each trace was parameterised with the *ShapeAnalysis* module. The maximum trace extent defined microbioerosion patterns as < 1 mm and macrobioerosion patterns as > 1 mm, following the definition by Wisshak (2012). Bioerosion rates (including micro- and macrobioerosion; g m$^{-2}$ yr$^{-1}$) were calculated by multiplying the volume of bioerosion (cm$^3$) with the mean density of limestone/marble of 2.7 g cm$^{-3}$ (Schön, 2011) and expressing the result per surface area (m$^2$) per duration of exposure (years). These rates include the residual

internal micro- and macrobioerosion, since the volume of substrate removed by grazers (potentially also including micro- and macrobioerosion) was impossible to quantify without a reference to the original substrate surface.

## 2.3   Bioerosion inventory

Bioerosion ichnotaxa were identified following descriptions of Bromley (1970) and Bromley and D'Alessandro (1983, 1984, 1989). Ontogenetic stages of sponge borings were classified into putative growth phases A-E according to characterisations

in Bromley and D'Alessandro (1984).

## 2.4   Sponge spicule analysis and species identification

For the identification of the trace makers of *Entobia* cavity networks, sponge spicule preparations were made from dry sponge tissue. To extract the tissue from the equivalent positions as in micro-CT reconstruction, the blocks were fragmented with hammer and chisel, and tissue was carefully removed with a dissecting needle. For spicule preparations sponge tissue

was digested in 68 % concentrated nitric acid in test tubes in a heated sand bath (60-70°C) for about 2 h, then leaving the solutions in place over night without heat application. On the next day, acid-cleaned spicules were washed three times in distilled water and dehydrated three times in laboratory-grade ethanol, each wash occurring after centrifugation and pipetting off the supernatant, taking care not to accidentally remove spicules. Spicules were then mounted for scanning electron microscopy (SEM) by drying aliquots of re-suspended spicules directly on SEM stubs, followed by sputter-coating with

gold, and analysis with the SEM (VEGA3, TESCAN). In some cases, colour of the dry tissue helped with classification efforts (e.g., Christomanos and Norton, 1974), but in most cases species identification had to rely exclusively on spicules.



We mostly referred to descriptions of Mediterranean bioeroding sponges by Rosell and Uriz (2002). Tylostyle measurements were obtained from 20 spicules per specimen. Spirasters and amphiasters were scarce and often broken, so that only five microscleres were measured where possible. Measurements were carried out using ImageJ v.1.48 (Rasband, 1997-2015).

**3 Results**

**3.1 Bioerosion traces**

Analysis of bioerosion patterns in the experimental blocks revealed ten different ichnotaxa, eight of which were attributed to the activity of excavating sponges and two to polychaete worms (Table 2). In respect to general patterns, the boring intensity distinctly increased with exposure time (Fig. 2-5). In blocks deployed for 2 to 5 years, superficial cavity networks were
observed (Fig. 2-3). From 7 years onward, extensive three-dimensional networks had developed (Fig. 4-5). Diversity increased over time as well, with blocks containing more than one ichnospecies after 5 years (Table 2).

In the two blocks deployed for 1 year no macroborings were detected (Fig. 2A-B). The first distinct sponge boring was observed in the 2-year block (Fig. 2C). The trace was characterised by cylindrical chambers (about 1.4-5.1 mm in length, 0.7-1.3 mm in width) that were arranged in long, sublinear chains that coalesced in cross-, T- or L-shape, which is
characteristic for *Entobia cateniformis* Bromley & D'Alessandro, 1984 in the late ontogenetic growth phase C. This system formed one tier in about 0.1-0.2 mm depth in the block parallel to the external surface, extending through the entire block.

In the 3-year block, an early-stage of *Entobia megastoma* (Fischer, 1868) was found. It was composed of a sublinear gallery (about 50 x 4 mm in size) with hand-like extensions (phase A-B; Fig. 2D). In the two 4-year blocks no assignable macrobioerosion patterns occurred (Fig. 2E-F).

In the 5-year block, a well-developed *Entobia geometrica* Bromley & D'Alessandro, 1984 cavity network was detected (Fig. 3A). The trace consisted of subrectangular to subtriangular flattened chambers with rounded corners (about 3.5-9.9 mm in diameter) that were aligned in weakly developed rows (phase D). The system extended in one tier throughout the entire surface of the block, parallel to the substrate surface in about 0.2-0.5 mm depth. In addition, one early-stage and two well-developed specimens of the polychaete bioerosion traces *Caulostrepsis* isp. penetrated from the upper surface into the block
(Fig. 3A1-3).

In the 6-year block, a well-developed network of *Entobia* cf. *ovula* Bromley & D'Alessandro, 1984 occurred (Fig. 3B). The trace was characterised by globose-ovoid to sub-prismatic chambers (about 0.7-1.5 mm in diameter) that were arranged in a crowded boxwork pattern (phase D). The entire system had a diameter of 40-50 mm and was arranged in one tier parallel to the external substrate surface in about 0.2-0.6 mm depth.

In the 7-year block, the bioerosion intensity strongly varied within the three subsamples (Fig. 4A-C). In the first subsample only an early-stage sponge boring was found that resembled *Entobia mammilata* Bromley & D'Alessandro, 1984 in growth phase A-B (Fig. 4A). In the second subsample, two separate specimens of *E. megastoma* and one of *E. mammilata* occurred




(Fig. 4B). The first specimen of *E. megastoma* was composed of subcylindrical galleries that formed a three-dimensional system (30 x 40 mm) and penetrated about 1.5 cm into the substrate (phase B-C; Fig. 4B1). The second specimen of *E. megastoma* was connected with *E. mammilata* (Fig. 4B2). This specimen of *E. megastoma* appeared to be an earlier growth stage than the other specimens in the same block (phase B). *Entobia mammilata* formed chains of turnip-shaped chambers

(about 1.3-1.7 mm; in phase B). The galleries extended over an area of about 42 x 40 mm and were arranged in two tiers in about 10 mm depth of the block. The third subsample was crowded with bioerosion traces that were identified as two entobian ichnospecies (Fig. 4C). Here, a well-developed network of *E. mammilata* was composed of clusters of tubercle-like chambers in about 1.2-1.4 mm of the block (phase D; Fig. 4C1). The network extended over an area of 30 x 90 mm and had penetrated the entire depth of the block (18 mm). In addition, several juvenile specimens of *E. megastoma* occurred. The

largest specimen was 25 x 50 mm in size and had a subcylindrical gallery of about 3 mm in diameter with long-exploratory threads (phase B; Fig. 4C2).

The 8-year block showed a dense network of shallow bioerosion (Fig. 5A), which was composed of three different entobians, dominated by a large network of *E.* cf. *ovula* clusters that were interconnected by long galleries in growth phase D. The largest cluster was about 25 x 40 mm in size and extended in about 0.2-0.6 mm depth of the block in one tier parallel

to the external substrate surface (Fig. 5A1). In addition, four large chambers of about 6 mm in diameter being connected by numerous exploratory threads were found that resembled *E. magna* Bromley & D'Alessandro, 1989 in phase B (Fig. 5A2). The third entobian was a specimen of *Entobia* cf. *parva* Bromley & D'Alessandro, 1989, which was identified through its compact boxwork of densely distributed, inflated chambers of about 0.2-0.6 mm that were arranged in a 15 x 40 mm tier in about 10 mm depth of the block (phase D; Fig. 5A3).

In the 8- to 9-year block, an advanced network of *E. megastoma* and several scattered early-stage galleries co-occurred (Fig. 5B). The 1-4 mm, subcylindrical galleries of the advanced specimen extended over an area of about 50 x 90 mm and were distributed in parallel to the block surface (phase B). The smaller specimens were about 10-15 mm in size and formed characteristic hand-like cavities (phase A).

The 14-year block was extensively bioeroded and entobians difficult to separate (Fig. 5C). Several specimens of *Entobia* cf.

*cretacea* Bromley, 1970 exhibited large polygonal chambers of about 1.3-4.2 mm in diameter, being connected by numerous canals, and with the entobians extending through the entire depth of the block (18 mm; Fig. 5C1-2). Bromley (1970) did not define distinct phases for *E. cretacea*, but we considered the detected traces to reflect phase C-D. Apart from entobians, three galleries of the worm boring *Trypanites* isp. extended as slim sack-like cavities into the blocks (Fig. 5C3-5). The software initially attributed these to the surrounding *E. cretacea* cavity network and the worm borings were afterwards separated

manually from the sponge borings.



## 3.2 Identification of boring sponges

We recognised spicules of five clionaid bioeroding sponge species that could be assigned as trace makers of eight different entobians (Fig. 6, Table 2, Table S1). However, not from every cavity network spicules could be extracted, and other, rare, broken or non-diagnostic spicules occurred that could not conclusively be matched to smaller entobians or used to identify

bioeroding sponges at species level (these possibly included spicules from the genera *Pione*, *Cliothosa*, *Spirastrella*, *Siphonodictyon,* and *Thoosa*).

*Cliona schmidtii* (Ridley, 1881) was recognised as the trace maker of *E. cateniformis* and *E. ovula* in the 2-year and the 8-year blocks (Fig. 6A-B, Table 2). Even when dry, the sponge tissue retained its characteristic purple colour that was an immediate indicator for *C. schmidtii*. Tylostyles were 170-275 µm in length and 3-8 µm in thickness, and had a slightly bent

shaft. A spectrum of spiraster sizes was observed: We distinguished (i) relatively thin and long spirasters with small spines distributed along the convex sides of the helical shaft (axis 57-84 µm in length and 1-3 µm in thickness), (ii) relatively short and thick spirasters with conical spines (axis 25-52 µm in length and 2-3 µm in thickness), and (iii) short, thick amphiaster-like spirasters (axis 17-21 µm in length and 3-7 µm in thickness).

Two morphospecies of *Cliona* cf. *celata* Grant, 1826 were distinguished as trace makers of two different entobians. *Cliona*

cf. *celata* 1 was trace maker of *E. geometrica* in the 5- year block (Fig. 6C, Table 2), and *Cliona* cf. *celata* 2 was matched to *E. megastoma* in the 7-year block (Fig. 6D-F, Table 2). Both morphospecies had brown dry sponge tissue and exclusively tylostyles. Tylostyles of *C.* cf. *celata* 1 were 243-373 µm in length and 6-13 µm in thickness, were comparatively robust with well-formed tyles that were occasionally subterminal, and had mostly straight or subtly bent shafts. Tylostyles of *C.* cf. *celata* 2 were comparatively slim, often with subterminal, occasionally multiple tyles, and tyles could be strongly displaced

or weakly pronounced, and occasional near-stylar modifications occurred. The tylostyle shafts of *C.* cf. *celata* 2 were occasionally flexuous, being on 208-369 µm in length, but only about 3-9 µm in thickness.

Based on scarce spirasters *Cliona* cf. *viridis* (Schmidt, 1862) was tentatively identified as trace maker of *E. ovula* and *E. mammilata* in the 6- and 7-year blocks (Fig. 6G-H, Table 2). The colour of the dry sponge tissue was brown. Tylostyles were 217-394 µm in length and about 3-9 µm in thickness, fusiform, slightly bent and had round, oval or subterminal tyles. Two

types of spirasters were distinguished: (i) straight, 20-44 µm in length and 1-2 µm in thickness, with relatively long spines that were mainly clustered at the ends of the shaft and (ii) helical, 18 µm in length and 1 µm in thickness, with small spines.

*Cliona rhodensis* Rützler & Bromley, 1981 was identified as trace maker of *E. magna* in the 8-year block (Fig. 6I, Table 2). The colour of the dry sponge tissue was brown. Tylostyles were near straight and fusiform, 245-356 µm in length and about 6-10 µm in thickness, and had distinct tyles. Only one complete, undamaged spiraster was found, which was 23 µm in length

and 1 µm in thickness, and had discrete, relatively long spines slightly recurving at their tips.





### 3.3 Bioerosion intensity and rates

Quantification of bioerosion in the experimental blocks revealed that in blocks deployed for 1 to 4 years only small volumes of substrate were removed by bioerosion (<1 %; Fig. 7A, Table S2). In the 5- to 8-year blocks, the volume of bioerosion increased to 4-9 %. The highest bioerosion intensity was measured in the block that was deployed for 14 years (30 %).

With respect to the proportional contribution of the different groups of bioeroding organisms to total bioerosion, the largest part of bioerosion in the 1- and 4-year blocks was represented by microbioerosion and unassignable macrobioerosion patterns (Fig. 7B). In all other blocks, bioerosion was predominantly produced by boring sponges. Only in the 5- and 14-year blocks complementary worm bioerosion was observed, but it contributed less than 1 % to the total volume of bioerosion.

Analogous to the gradual increase in total volume of bioerosion, total bioerosion rates (i.e. residual micro- and macrobioerosion) increased with exposure time (Fig. 7C, Table S3). Lowest values were found in the 1- to 4-year blocks (1.5-85 g m$^{-2}$ yr$^{-1}$), except in the 2-year block, where bioerosion rates tallied with 224 g m$^{-2}$ yr$^{-1}$. In the 5- to 7-year blocks bioerosion was elevated compared to blocks retrieved after shorter periods and reached values of 308-648 g m$^{-2}$ yr$^{-1}$. However, in 7- and 8 to 9-year blocks lower values of 53 and 62 g m$^{-2}$ yr$^{-1}$ were observed. Highest bioerosion rates were measured in the 14-year block, resulting in a maximum value of 900 g m$^{-2}$ yr$^{-1}$. Overall, observed bioerosion patterns suggest that not only the bioerosion intensity, i.e. the absolute volume removed by bioerosion, but also the bioerosion rates, i.e. bioerosion normalised to a time span of one year, increased with time of exposure. An additional statistical evaluation was not considered feasible, however, due to the limited amount of available blocks suitable for micro-CT analysis.

## 4    Discussion

### 4.1 Micro-CT as a tool for the visualisation and quantification of internal macrobioerosion

Methods used in this study represent a new approach and perspective for precise and automatic differentiation and quantification of internal structures of bioerosion, and they can be employed for similar aspects in biogeoscience research. Previously, evaluation of internal bioerosion by tomographic analysis was restricted to two-dimensional image analysis of consecutive layers (e.g., Becker and Reaka-Kudla, 1997; Hassan, 1998; Sammarco and Risk, 1990; Schönberg, 2001), and only recently included also three-dimensional measurement tools (Crook et al., 2013). Above methods are suitable for comparatively simple bioerosion structures, but show clear limitations in the differentiation of complex cavity networks. In our study we took advantage of program algorithms that help to distinguish different traces. This function needs to be manually revised, however, as it does not automatically identify ichnotaxa according to morphological differences. This becomes especially clear where morphological distinct traces were identified as fused to one connected cavity network (Fig. 4B2, 5A), an effect likely caused by insufficient separation of traces due to different generations of endoliths overprinting earlier borings or galleries of bioeroding sponges that can closely mingle with other species or individuals (unlike observations in Bromley and Tendal, 1973).



Another restriction of micro-CT analysis is the correlation of sample size and spatial resolution. Based on the sample dimensions of 10 x 10 cm spatial resolutions of 72 µm were possible, which allowed preserving large cavity networks, as well as capturing thin exploratory threads of the sponges. However, where connecting galleries were near or below this resolution, occasionally different units of the same trace were split into apparently different specimens (Fig. 4C). Neither

was the present resolution high enough to capture smaller microborings. Subsequent manual separation or joining of specimens needs to be assessed case-by-case. That is comparatively easy and feasible for cylindrical borings such as *Trypanites* isp. (Fig. 5C3-5), but for complex cavity networks, such as sponge borings, this can potentially become very time-consuming.

Apart from microborings and thin exploratory threads, a small proportion of deeper borings was also neglected due to the

digital cropping of the blocks. In addition, some proportion of bioerosion became unavailable for quantification when grazers simultaneously removed surface layers and micro- as well as macrobioerosion traces within them. This substrate loss cannot be quantified without a reference that indicates the original thickness of the experimental blocks. What was actually measured, hence, is the residual internal bioerosion. Particularly in those blocks that were exposed long-term, bioerosion rates thus were somewhat underestimated, in turn implying that the observed increase in bioerosion rates can be expected to

be even more pronounced for total bioerosion.

## 4.2 The macrobioeroder community

We distinguished *C.* cf. *viridis*, *C. schmidtii*, *C.* cf. *celata* morphospecies 1 and 2, and *C. rhodensis* as main producers of the observed bioerosion traces. Apart from these, we assume that other sponges bioeroded the blocks as well, as evidenced by further rare spicules and by few traces that may have differed from the above, but from which no spicules could be extracted.

This possibly includes the locally common boring sponges *Pione vastifica* (Hancock, 1849), and *Cliothosa hancocki* (Topsent, 1888), which Rützler and Bromley (1981) and Bromley et al. (1990) reported from Rhodes.

Bioeroding sponge distributions and abundances, as well as their bioerosion rates, are dependent on environmental parameters such as water flow, nutrients, salinity, temperature, and light (see Schönberg, 2008 for a review). Especially *C. viridis* and *C. celata* are very characteristic for the Mediterranean and known to be among the most abundant and destructive

sponges in the Mediterranean Sea (e.g., Calcinai et al., 2011; Rosell et al., 1999). Both species, however, are difficult to identify and are members of species complexes that encompass very similar, but taxonomically different species, which means that earlier accounts on their biology may refer to more than one species (e.g., Leal et al., 2015; Xavier et al., 2010). This may also be the case for *C. schmidtii*, because specimens from different sample sites can have somewhat different spicule morphologies (Schönberg, pers. obs.). However, *C. schmidtii* and *C. rhodensis* were the presently best confirmed

species, but little is known about their ecological requirements: Both can be found in light and shade, and occur in moderately clear water and may avoid sedimentation (Rützler and Bromley, 1981; Carballo et al., 1994; Corriero et al., 2000), but it is not known what else characterises their ecological niches. To draw conclusions about environmental




conditions from our data is difficult, because blocks were from different depths and current regimes, and the lack of replication prevented us from matching species distributions with environmental conditions.

Although on the surrounding sea floor also boring bivalves and worms (sipunculans, polychaetes) were reported to be very common, our experimental blocks were dominated by boring sponges. We assume that this dominance of sponges is an

effect of the high spatial variability of macroborers and that locally sponges were most prevalent and determined the predominant larval supply. This assumption is in good agreement with results from other long-term experiments in the tropics, where macrobioerosion strongly varied with water depth and nutrient supply (Carreiro-Silva and McClanahan, 2012; Kiene and Hutchings, 1992, 1994; Osorno et al., 2005; Pari et al., 1998, 2002; Peyrot-Clausade et al., 1995). The low occurrence of polychaete borings in the blocks may be also explained by the fact that polychaetes have comparatively short

life spans and that their vacant burrows became inhabited and overprinted by newly settled larvae of other boring species (Hutchings et al., 1992). To draw a direct comparison between bioerosion rates from these studies and our experiment, however, is difficult, because these experiments were based on coral substrates and assessed via image analysis.

### 4.3 Palaeoenvironmental implications

In the fossil record, sponge borings are preserved as trace fossils in calcareous hard substrates such as rocky shores,

hardgrounds, or shells providing important information about palaeoenvironmental conditions (Wilson, 2007). Whereas recent boring sponges can be identified by their spicules or other morphological and molecular characters, the description of fossil entobians mostly relies on the morphological characterisation of their bioerosion traces. This is, because the preservation potential of the boring is much higher than that of the siliceous spicules, which are only rarely preserved within the borings (e.g., Blissett et al., 2006; Bromley and Schönberg, 2008; Reitner and Keupp, 1991). Thus, in order to allow

conclusions about past environmental conditions drawn in an actualistic approach from the ecophysiology studied in extant sponges, traces of recent sponges need to be matched with the sponge spicule record, so that deductions about fossil trace makers can be made where a close morphological resemblance is found between fossil and recent traces.

This study is one of the few that allowed matching sponge borings (bioerosion traces) and boring sponges (trace makers) by combining micro-CT with spicule analysis, providing respective information for five species of recent sponges. This

information furthers an earlier detailed inventory of Mediterranean sponge borings and boring sponges by Bromley and D'Alessandro (1989). These authors were also able to obtain some clear matches between sponge species and ichnospecies, but in accordance with our results they also showed that one sponge species can produce different traces. Especially *C. celata* was described several times as producing traces that vary morphologically (Bromley and D'Alessandro, 1989; De Groot, 1977), *E. geometrica* and *E. megastoma*, even in the same type of substrate. However, as mentioned above, *C. celata*

is not a single species, but a species complex of several morphologically indistinct species, and a proper separation of species presently relies on molecular taxonomy (De Paula et al., 2012; Xavier et al., 2010). While spicular morphology can be variable with different environmental conditions (Bavestrello et al., 1993; Hoeksema, 1983; Rosell and Uriz, 1991), we



recognised subtle but consistent differences in spicule morphology between *C.* cf. *celata* 1 (producing *E. geometrica*) and *C.* cf. *celata* 2 (producing *E. megastoma*), and in this case different traces may in fact represent different trace makers. A similar situation may be the case for the two borings found in the other difficult species, *C.* cf. *viridis* (*E. ovula* and *E. mammilata*), but as present samples were not preserved for molecular analysis, we cannot confirm or reject this assumption.

In contrast to *C. celata*, we distinguished two different entobians for *C. schmidtii*: *E. cateniformis* and *E.* cf. *ovula*. Bromley and D'Alessandro (1989) also identified *C. schmidtii* as trace maker of *E. ovula*, but in their study *E. cateniformis* was produced by *P. vastifica* (formerly *C. vastifica*). We furthermore identified *C.* cf. *viridis* as trace maker for *E. ovula* and *E. mammilata*, which led to the conclusion that not only the same species can produce different traces, but the same trace can be produced by different species – which is not a surprise but in accordance to basic ichnological principles (e.g., Bromley

and Fürsich, 1980). Our results thus also agree with the assumption of Bromley and D'Alessandro (1984) that the morphology of entobians can strongly vary with the nature and structure of the substrate, the quality of the surrounding environment, the proximity of other endoliths, the species of the boring sponge, and the ontogeny of the borer. However, we lack comparative data as all presently available research on the correlation of sponge borings and boring sponges was carried out in Mediterranean Sea. Further studies are needed in order to ascertain whether the observed correlations also apply to

different biogeographic realms, and to better understand the application and limitations of sponge bioerosion traces as palaeoenvironmental indicators.

## 4.4 Long-term succession of boring sponges: Are 14 years long enough to develop equilibrium communities in a warm-temperate environment?

This study provides one of the longest records for a bioerosion experiment, and is one of the few available from the

Mediterranean. It included quantitative analyses and observations on succession dynamics of macrobioeroders, all of which represents vital information to assess the impact of macrobioerosion on marine carbonate environments. Using the approach of micro-CT in combination with sponge spicule analysis revealed that during the experiment the blocks were predominantly bioeroded by sponges. The present study from a warm-temperate habitat confirms findings from tropical coral reefs that sponges require a few years to colonise newly available substrates and may only form larger infestations after more than 7

years (e.g., Kiene and Hutchings, 1992, 1994). Based on the present results and taking earlier macroscopic observations of the same blocks into account (Bromley et al., 1990), we recognised two developmental phases during our experiment: (1) an early community stage with initial sponge bioerosion and comparatively low macrobioerosion rates between years 1 and 5 and (2) an intermediate stage starting in year 6 or 7 when boring sponges become firmly established and bioerosion rates increase. Especially in the latter phase, our material displayed much variability, which we expect to decrease at later stages,

when a stable bioeroding community has established. Across the early and intermediate stages, bioerosion rates increased over time, but can be expected to slow down to a relatively stable rate when equilibrium conditions are eventually reached. Based on the present dataset, a sound prediction about the onset of these equilibrium conditions is not feasible, and would





require even longer exposures combined with a higher number of replicates, and such experiments would be necessary to determine whether bioerosion rates peak at an equilibrium plateau, or, more likely, that highest bioerosion rates are reached in the intermediate stage of colonisation, when substrate is not yet limited and competition not yet restricting further growth. The analysis of ontogenetic stages of the observed entobians, nevertheless, suggests a distinct development of sponge

bioerosion over time. Having settled down on suitable substrates and finding ample space, boring sponges can rapidly mature, preferentially by lateral extension. With more specimens colonising the blocks, boring sponges formed increasingly three-dimensional patterns, but less developed ontogenetic stages. This is in good agreement with observations by Rützler (1975), who showed that with sufficient space and little competition, sponge borings mostly spread laterally and progressively bore vertically when the substrate edges are reached or lateral spreading is compromised by neighbouring

competitors or other limitations. The presence of an increasing number of vertical borings during the intermediate stage in our experiment could indicate the gradual onset of a saturation phase. Similar observations with macrobioerosion rates proceeding at lower rates when free substrates became scarce and crowded with borings were demonstrated during long-term studies from the tropical realm (Carreiro-Silva and McClanahan, 2012; Kiene and Hutchings, 1994; Lescinsky et al., 2008). At present, however, we cannot conclude how much time is needed in warm-temperate environments to reach equilibrium

phases. This again underlines the necessity for further long-term studies, especially outside the tropical realm.

**5 Conclusions**

This study presents the first record on long-term bioerosion from the warm-temperate realm and showcases the use of micro-CT to study internal bioerosion. In contrast to previous studies, experimental blocks were almost exclusively colonised by

boring sponges, while only few worm and no bivalve borings were observed. Analyses of bioerosion traces and rates suggested an early community development stage during the first 5 years of exposure, where first boring sponges settle, but yield low rates of macrobioerosion, and an intermediate stage commencing in years 6 to 7 when boring sponges matured and bioerosion rates increase. After 14 years, 30 % of the block volumes were occupied by boring sponges. Analysis of ontogenetic stages of sponge borings suggested that successful settlement of boring sponges is strongly dependent on the

availability of space and on competition by other bioeroders. A high spatial variability in macrobioerosion prohibited clear conclusions about the onset of macrobioerosion equilibrium conditions. More long-term experiments are needed in order to identify equilibrium conditions and to assess the impact of macrobioerosion in different biogeographic realms.

**Acknowledgements**

We thank Nicol Mahnken (Senckenberg am Meer) for technical support and Ines Pyko (Erlangen) for her help in inventorying the experimental blocks. This study was funded by the Deutsche Forschungsgemeinschaft (DFG Wi 3754/2-1). This paper is published with the permission of the Director of the Institute of Geology and Mineral Exploration, Athens, Greece.



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





Table 1. Metadata of experimental blocks deployed during the experiment and analysed via micro-computed tomography (E = east, W = west, Ma = marble, Mi = micritic limestone).

| Exposure (yrs) | Site | Coast | Water depth (m) | Deployment | Recovery | Block lithology |
|---|---|---|---|---|---|---|
| 1 | Lindos | E | 3-8 | 21.10.1982 | 25 October 1983 | Ma |
| 1 | Pyrgos | W | 3-6 | 20.10.1982 | 26 October 1983 | Ma |
| 2 | Pyrgos | W | 8 | 26.10.1983 | 15 October 1985 | Ma |
| 3 | Ladiko | E | 3-4 | 28.10.1983 | 22 October 1986 | Ma |
| 4 | Lindos | E | 16-17 | 19.10.1982 | 19 October 1986 | Mi |
| 4 | Ladiko | E | 3.5 | 28.10.1983 | October 1987 | Ma |
| 5 | Ladiko | E | 3 | 28.10.1983 | 14 October 1988 | Mi |
| 6 | Lindos | E | 12 | 18.10.1985 | 18 October 1991 | Ma |
| 7 * | Kolimbia | E | 3 | 17.10.1982 | 13 October 1989 | Ma |
| 8 | Lindos | E | 12 | 18.10.1985 | 1993 | Ma |
| 8-9 | Pyrgos | W | 3 or 8 m | 20.10.1982/83 | 16 October 1991 | Ma |
| 14 | Pyrgos | W | 7 | 20.10.1982 | 1996 | Ma |

* three replicates scanned



Table 2. Inventory of bioerosion traces. Sponge borings (trace = *Entobia* ispp.) and boring sponges (trace maker = *Cliona* spp.) in the scanned experimental blocks were assigned via spicule analysis. Ontogenetic phases of sponge boring traces were determined according to Bromley and D'Alessandro (1984) with most mature stages in brackets. Measurements of sponge spicules are given as ranges, with length before widths, and means in parenthesis. Number of tylostyles n = 20, number of spirasters n = 5, unless otherwise indicated in square brackets.

| Exposure (yrs) | Boring | | Phase | Producer | | Tylostyles (µm) | Spiraster (µm) |
|---|---|---|---|---|---|---|---|
| 2 | *Entobia cateniformis* | Fig. 2C | C (D) | *Cliona schmidtii* | Fig. 6A | 170-247 (198) x 3-8 (5) | a) 57-71 (66) x 2-3 (2)<br><br>b) 28-52 (36) x 2-3 (3)<br>c) 17 x 3 [1] |
| 3 | *Entobia megastoma* | Fig. 2D | A-B (D) | Sponge | - | - | - |
| 5 | *Entobia geometrica* | Fig. 3A | D (D) | *Cliona* cf. *celata* 1 | Fig. 6C | 243-373 (320) x 6-13 (10) | - |
| | *Caulostrepsis* isp. | Fig. 3A1-3 | | Worm | | | |
| 6 | *Entobia* cf. *ovula* | Fig. 3B | D (D) | *Cliona* cf. *viridis* | Fig. 6H | 217-356 (273) x 3-6 (4) | 20-32 x 1 [2] |
| 7 | *Entobia* cf. *mammilata* | Fig. 4A | A-B (E) | Sponge | - | - | - |
| 7 | *Entobia megastoma* | Fig. 4B1 | B-C (D) | *Cliona* cf. *celata* 2 | Fig. 6D | 218-369 (316) x 4-8 (6) | - |
| | *Entobia megastoma* | Fig. 4B2 | B (D) | *Cliona* cf. *celata* 2 | Fig. 6E | 208-339 (283) x 3-9 (6) | - |
| | *Entobia mammilata* | Fig. 4B2 | B (E) | Sponge | - | - | - |
| 7 | *Entobia mammilata* | Fig. 4C1 | D (E) | *Cliona* cf. *viridis* | Fig. 6G | 234-394 (324) x 4-9 (7) | a) 30-44 (35) x 1-2 (1)<br>b) 18 x 1 [1] |
| | *Entobia megastoma* | Fig. 4C2 | B (D) | *Cliona* cf. *celata* 2 | Fig. 6F | 235-358 (321) x 5-9 (7) | - |
| 8 | *Entobia* cf. *ovula* | Fig. 5A1 | D (D) | *Cliona* cf. *schmidtii* | Fig. 6B | 20) x 3-6 (5) | a) 58-84 (75) x 1-2 (2)<br>b) 25-46 (35) x 2 (2)<br>c) 21 x 7 [1] |
| | *Entobia* cf. *magna* | Fig. 5A2 | B (D) | *Cliona rhodensis* | Fig. 6I | 245-356 (289) x 6-10 (8) | 23 x 1 [1] |
| | *Entobia* cf. *parva* | Fig. 5A3 | D (D) | Sponge | - | - | - |
| 8-9 | *Entobia megastoma* | Fig. 5B | B (D) | Sponge | - | - | - |
| 14 | *Entobia* cf. *cretacea* | Fig. 5C | C-D (D) | Sponge | - | - | - |
| | *Trypanites* isp. | Fig. 5C3-5 | | Worm | | | |




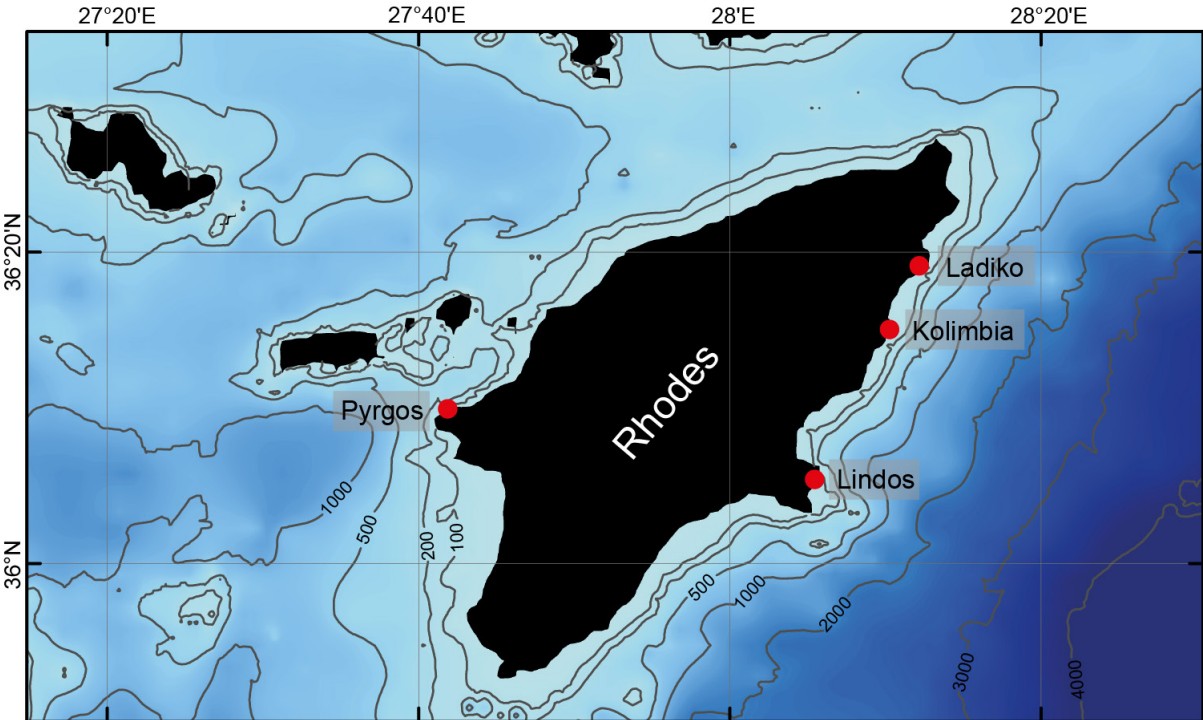

Figure 1. Location of the long-term experiment at Rhodes, Greece, in the Eastern Mediterranean Sea. Between 1982 and
5    1996, experimental blocks were deployed at four sites on the east and west coast of Rhodes in 3 to 17 m water depth.
Bathymetric data were derived from the EMODnet Portal for Bathymetry (EMODnet, 2015).





Figure 2. Micro-computed tomographic visualisation of bioerosion traces in experimental blocks deployed for 1-4 years. (A-B) 1-year blocks showed no macrobioerosion traces. First bioerosion traces were observed (C) in the 2-year block with *Entobia cateniformis* and (D) in the 3-year block with *E. megastoma*. (E-F) The 4-year blocks showed no distinct macrobioerosion traces (size of blocks = 90 x 90 x 18 mm).





Figure 3. Micro-computed tomographic visualisation of bioerosion traces in experimental blocks deployed for 5-6 years. (A) The 5-year block showed a network of *Entobia geometrica* and specimens of *Caulostrepsis* isp. (A1-A3). (B) The 6-year block showed a cluster of *E.* cf. *ovula* (scales A1-A3 = 2 mm, size of blocks = 90 x 90 x 18 mm).





Figure 4. Micro-computed tomographic visualisation of bioerosion traces in three subsamples of the 7-year block. (A) First subsample of the block with *Entobia* cf. *mammilata*, B) second subsample of the block showing two specimens of *E. megastoma* (B1 and large chambers B2) and *E. mammilata* (small chambers B2), and C) third subsample of the block with *E. mammilata* (C1) and *E. megastoma* (C2) (scales = 10 mm, size of blocks = 90 x 90 x 18 mm).





Figure 5. Micro-computed tomographic visualisation of bioerosion traces in experimental blocks deployed for 8-14 years. (A) 8-year block with *Entobia* cf. *ovula* (A1), *E. magna* (A2), and *E.* cf. *parva* (A3), B) 8-9-year block with *E. megastoma*, and C) 14-year block with *E.* cf. *cretacea* (C1-C2) and *Trypanites* isp. (C3-C5) (scales A1-A3, C1-C2 = 5 mm, scales C3-C5 = 1 mm, size of blocks = 90 x 90 x 18 mm).





Figure 6. Spicules of boring sponges extracted from sponge tissue preserved in cavity networks in the scanned blocks. (A-B) Tylostyles and spirasters of *Cliona schmidtii*, (C) tylostyles of *Cliona* cf. *celata* 1, (D-F) tylostyles of *Cliona* cf. *celata* 2, (G-H) tylostyles and spirasters of *Cliona* cf. *viridis*, (I) tylostyles and spirasters of *Cliona rhodensis* (Fig. 5A2) (scales: tylostyles = 50 µm, spirasters = 20 µm).



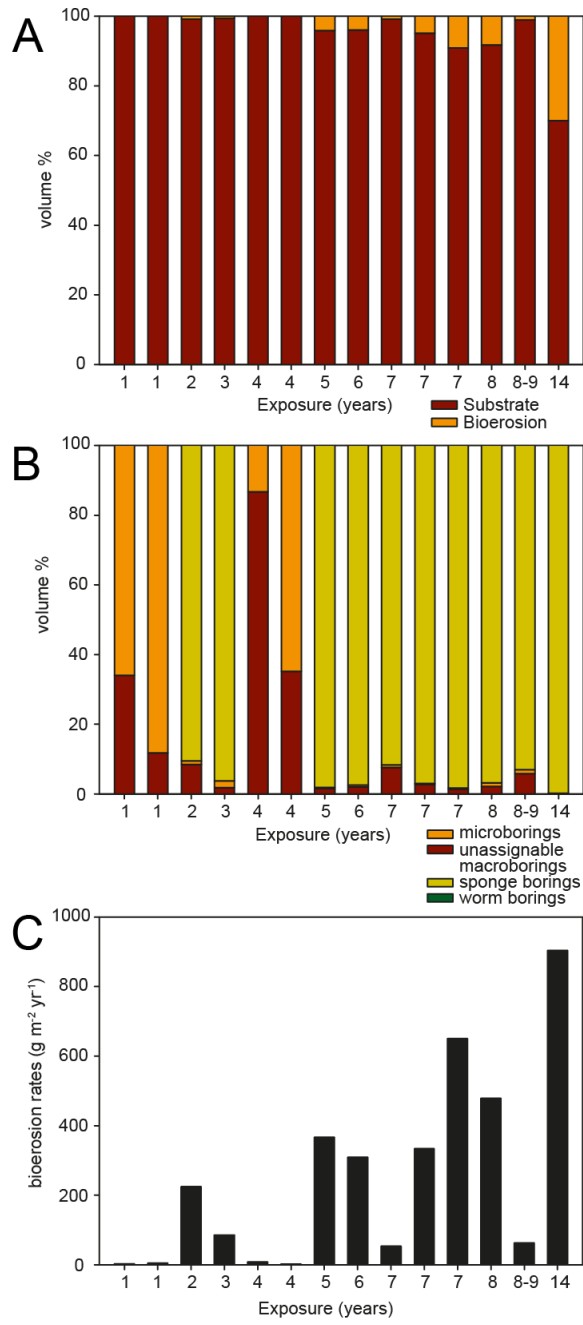

Figure 7. (A) Proportion of retained substrate vs. internal bioerosion in the scanned blocks, (B) proportional contribution of the different groups of boring organisms to the internal bioerosion in A, and (C) internal bioerosion rates (g m$^{-2}$ yr$^{-1}$)

5    measured in experimental substrates during the experiment.