# Peer review of "Long-term macrobioerosion in the Mediterranean Sea assessed by micro-computed tomography"

_Biogeosciences, 2015_

## Referee Comment (RC1) · W. Kiene (Referee) · 10 Feb 2016

This is an excellent study, which describes a valuable technique for characterizing and quantifying bioeroding communities in hard substrates. I commend the authors for doing this and have added substantially to the understanding of the colonization of boring communities and the excavations they make in carbonate surfaces. I have the following comments and suggestions:

Experimental design: It is not clear what the original sizes of the experimental blocks were. Surface area of the blocks is said to be 11 x 11 cm or more. Blocks have three dimensions. How were they deployed on the sea floor? The different lithologies of the blocks and variations in their deployment and size likely had significant influences on

the results (such as the lack of colonization in the 4-year collection but 2 and 3 year blocks were colonized). This is referred to on Page 10, Lines 1-2, but some indication of these variations would be helpful in the methods. On Page 11, Line 26, it is indicated that the blocks came from a previous study by Bromley. Please refer to this in the methods.

Page 3, Line 21: What are "the following called blocks"?

Page 4, Line 3: Blocks were digitally cropped for analysis. Were these the dimensions used to quantify area for rates of bioerosion indicated in Line 14, or was the cut size of the blocks used? The physical cutting and digital cropping does not consider the original surface area on which the bioeroders established. This may not be too important though, since the characterization and volume of borings are the main result, and the bioerosion rates calculated are not critical to the conclusions.

Page 4, Line 24: Were the fragmented blocks the same ones that were also digitally scanned?

Page 8, Line 31: Suggest changing "closely mingle" to "live closely together" or cohabitate, if this is what you mean. Are you suggesting that different sponge species can occupy the same gallery?

Page 10, Line 11: Hutchings et al didn't use image analysis, but quantified erosion on sections cut through experimental blocks and point-counting erosion areas under a microscope.

Page 10, Line 17: Remove the "," after "This is"

Page 11, Line 30: Does a "stable bioeroding community" establish if the surface of a bored substrate is eroded by grazing or other action? If a surface is eroded by grazing, new uncolonized substrate would be continually exposed. The intensity of this grazing and subsequent exposure could also vary substantially in space and time. Such factors would need to be isolated to conclude what a "stable" community of borers or rate of

[Figure]

bioerosion looks like.

Table 1: Explain how the three replicates of the 7-year sample were made
[Figure]

---

## Referee Comment (RC2) · Anonymous Referee #2 · 18 Feb 2016

Long-term macrobioerosion in the Mediterranean Sea assessed by micro-computed tomography" Färber et al

Referee's suggestions

The study is very interesting and gives, for the first time, data about long-term marine bioerosion in temperate waters; the use of m-CT for quantifying bioerosion rates is really promising.

Page 3 line 3. 46 blocks of marble and limestone were used for the experiment and placed on the sea floor. You chose 20 blocks for your analysis. Could you detail the lithology of the chosen blocks?
[Figure]

Page 3 line 13, page 8 line 17. You cut blocks to a size of 10x10 cm. What is their thickness? What is the size of the blocks used for micro-CT? is it 90x90x18 mm as reported in the legends? How did you choose the parts of the blocks for micro-CT analysis?

Pag 6 line 23- Cliona schmidti. See comments for the legend of figure 3. line 24 and followings. When you quote the figure 6 with the spicules of the sponges, please quote also the corresponding figs of the biorosion traces got with m-CT.

Discussion page 8. Could you consider if the lithology of the blocks affect the results of the erosion rates?

page 9 lines 15-18. Please add comments about the dominance of sponges in the excavated blocks, discussing the succession phase of bioeroders. Sponges may pre-cede worms and bivalves in colonisation (e.g. Hutchings, 1986). This could be true only for tropical waters, and moreover, these data about successional phases refer to shorter time gaps. But in temperate areas bioerosion rates are slow (for Teredinidae is demonstrated). Is it for this reason that worms are scarce? Moreover, these blocks are small (10x10 cm). Likely, they moved and rolled on the bottom. Worms (with sin-gles openings) could have been smothered by mud, and the vacant holes occupied by sponges.

page 10 line 20-25. Is it your case? the different lithology of the blocks could have influenced also the erosion traces?

Figure 1. Show the Mediterranean sea and where Rhodes is. Figure 3. Could be possible that in B two distinct traces are presents? Apart E. ovula, there are other cylindrical chambers arranged in chains, very similar to fig. 2C (E. cateriformis). In fact, the erosion traces in 2C are attributed to C. schmidti, and the traces in figure. 3B to two distinct species: Cliona viridis and C. schmidti. The chambers arranged in chains in Fig. 3C could belong to C. schmidti and the others, globose-ovoid to C. viridis. So the sentence " Cliona schmidtii (Ridley, 1881) was recognised as the trace maker

[Figure]

of E. cateniformis and E. ovula " should be reconsidered.

---

## Author Response (AR1)

Dear Professor Kitazato,

We would like to thank you, Mr. William Kiene, and the anonymous reviewer for the very detailed and careful reviews.

Please find attached our reply to your and the reviewers' comments that were followed in most instances. Some minor corrections were also made during proofreading.

Sincerely yours,

Claudia Färber and coauthors

**Editor Comments**

*1) You used micritic limestones and marbles. Does all the experimental blocks originate from Rhodes Island?*

Response: According to Bromley et al. 1990 the blocks were surplus material of the Greek building industry and their provenance is unknown. However, in the present context we do not consider this to be disadvantageous or relevant for the experimental design.

*2) You found ecological succession of microboring by sponges. These sponge groups all live in Rhodes Island? If not, how many percentages of local sponges are repopulated on experimental blocks.*

Response: According to the editor's comment we added information regarding the abundance of boring sponges at Rhodes and the Eastern Mediterranean Sea:

Page 9, Line 17-21: "We distinguished *C.* cf. *viridis*, *C. schmidtii*, *C.* cf. *celata* morphospecies 1 and 2, and *C. rhodensis* as main producers of the observed bioerosion traces. When ignoring our second morphospecies of *C. celata*, the retrieved species represented 36 % of the diversity of the locally reported Clionaidae, or 50 % of the reported *Cliona* spp. for the Eastern Mediterranean (e.g. Evcen and Çinar, 2015; Vacelet et al., 2008). Apart from these, we assume that other sponges bioeroded the blocks as well, as evidenced by further rare spicules and by few traces that may have differed from the above, but from which no spicules could be extracted. This possibly includes *Cliothosa hancocki* (Topsent, 1888), which was reported together with *C. viridis* and *C. rhodensis* to be one of the most abundant sponges at Rhodes (Rützler and Bromley, 1981)."

**Review 1 (William Kiene)**

*1) Experimental design:* It is not clear what the original sizes of the experimental blocks were. Surface area of the blocks is said to be 11 x 11 cm or more. Blocks have three dimensions. How were they deployed on the sea floor? The different lithologies of the blocks and variations in their deployment and size likely had significant influences on the results (such as the lack of colonization in the 4-year collection but 2 and 3 year blocks were colonized). This is referred to on Page 10, Lines 1-2, but some

indication of these variations would be helpful in the methods. On Page 11, Line 26, it is indicated that the blocks came from a previous study by Bromley. Please refer to this in the methods.

Response: We noticed that both reviews expressed confusion about the experimental design. In order to clarify this, we added a brief summary of the original description of the experiment in Bromley et al., 1990, addressing questions about the size of the blocks, the deployment, and the preparation for micro-CT analysis. With respect to the lithology of the blocks, we agree with the reviewers that differences in lithology could be a source of bias for the succession of bioeroders. We used marble blocks throughout with the only exception of two micritic limestone blocks (one 4- and the 5-year block). The lithology of the blocks was examined in thin sections and is described in Table 1. We agree with reviewer 1 that the lack of bioerosion in the 4-year blocks is striking and might be an effect the micritic lithology of one of them. However, the other 4-year block (marble) shows no indication of macrobioerosion either, while in the micritic 5-year block intensive bioerosion was recorded. Hence, while we cannot entirely rule out an effect of the different lithologies, such an effect cannot be confirmed with the present data.

According to the reviewers' comments we modified chapters 2.1 and 2.2 of the Methods section as follows and added a short comment on lithological differences in the Discussion:

2.1 Experimental design (page 3, line 3-16): "The settlement experiment was carried out in the vicinity of four limestone cliffs (...) The original description of the experiment is provided in Bromley et al., 1990 and is summarised here as follows: Between 1982 and 1989, experimental blocks were deployed in water depths between 3 and 17 m. The blocks were of marble and micritic limestone, and respective lithologies were confirmed by petrographic thin sections (Table 1). Initially, smaller blocks (1-3 kg) were placed directly on the sea floor, but many of these were lost during winter storms in the first year. Accordingly, larger blocks were laid out (5-30 kg) and smaller blocks were tied securely to iron or plastic frames to increase their stability. However, except in the most turbulent sites, anchoring of blocks was avoided to simulate natural conditions as closely as possible. Between 1982 and 1996 each year some blocks were retrieved, all by skin diving and using floatation devices. Recovered blocks were rinsed in fresh water, and soft epiliths were removed. The blocks were subsequently photographed and dried."

2.2 Micro-computed tomography (page 3, line 18-21): "Internal bioerosion in the blocks was investigated by micro-computed tomographic analysis (micro-CT). From the whole inventory of 46 recovered blocks, twelve were chosen from different depths (3 to 17 m) and exposure times (1 to 14 years), all with a surface area of 11 x 11 cm or larger. Most of the blocks showed no evidence for having been dislocated during the experiment, however, some had slightly moved and for some blocks the information about the recovery date and/or water depth was incomplete (Table 1). This was reviewed case by case, and ultimately these blocks were included in the study. In order to yield a spatial resolution of about 70 µm, the chosen blocks were cut with a rock saw to a uniform surface area of 10 x 10 cm. The thickness of these blocks varied between 2 to 5 cm, but to keep mechanical destruction as small as possible, adjusting the thickness was avoided at this stage. The 7-year block was cut into three such blocks as replicates to obtain an impression of the spatial variability of bioerosion within a single block. From blocks that were large enough to be cut into several subsamples, replicates were randomly chosen. In this way, a total of 14 samples were produced for micro-CT analysis. Micro-CT scanning was carried out (...)"

4.2 The macrobioeroder community (page 10, line 3): "In addition to environmental parameters, sponge bioerosion is sensitive to substrate characteristics such as lithology, density, porosity, crystal size, and the presence of siliciclastic fragments (e.g., Calcinai et al., 2007). During the experiment mostly marble blocks were used, but also two micritic limestone blocks: one 4-year block that showed no indication of macrobioerosion (Fig. 2E), and the 5-year block that was intensely bioeroded (Fig. 3A). Based on the present material, substrate effects cannot be ruled out, but are difficult to identify and distinguish from environmental effects."

*2) Page 3, Line 21:* What are "the following called blocks"?

Response: This was meant to address that subsequently the cut samples were used for micro-CT analysis and no longer represented the original-sized "blocks". However, we recognise that this is unnecessary and omitted it.

*3) Page 4, Line 3:* Blocks were digitally cropped for analysis. Were these the dimensions used to quantify area for rates of bioerosion indicated in Line 14, or was the cut size of the blocks used? The physical cutting and digital cropping does not consider the original surface area on which the bioeroders established. This may not be too important though, since the characterization and volume of borings are the main result, and the bioerosion rates calculated are not critical to the conclusions.

Response: Bioerosion rates were calculated in relation to the surface of the digitally cropped blocks. The first step, the mechanical cutting of the blocks with a rock saw was predominantly required to reduce the sample size for higher spatial resolution. As mentioned by the reviewer, based on this method we could not refer to the original surface area on which the bioeroders established (cf. page 9, line 9-15). This would have been, of course, more precise and would also have allowed including the grazer impact, but would have required scanning the blocks before deployment as reference. This will be, of course, realised in future experiments.

*4) Page 4, Line 24:* Were the fragmented blocks the same ones that were also digitally scanned?

Response: Yes. In order to clarify this, we added "For the identification of the trace makers of *Entobia* cavity networks, sponge spicule preparations were made from dry sponge tissue preserved in the scanned blocks"

*5) Page 8, Line 31:* Suggest changing "closely mingle" to "live closely together" or cohabitate, if this is what you mean. Are you suggesting that different sponge species can occupy the same gallery?

Response: "Closely mingled" was meant to address that different specimens of boring sponges were closely inhabiting one substrate and almost merging. According to the reviewer's comment we modified the sentence as follows: "that were produced by species or individuals living closely together".

*6) Page 10, Line 11:* Hutchings et al didn't use image analysis, but quantified erosion on sections cut through experimental blocks and point-counting erosion areas under a microscope.

Response: According to the reviewer's comment we specified the applied methods: "(...), because these experiments were based on coral substrates and different quantification methods such as point-counting of sections (Kiene and Hutchings, 1992, 1994) and image analysis of sections (Carreiro-Silva and McClanahan, 2012; Osorno et al., 2005; Pari et al., 1998, 2002).

*7) Page 10, Line 17:* Remove the "," after "This is"

Response: Was changed accordingly.

*8) Page 11, Line 30:* Does a "stable bioeroding community" establish if the surface of a bored substrate is eroded by grazing or other action? If a surface is eroded by grazing, new uncolonized substrate would be continually exposed. The intensity of this grazing and subsequent exposure could also vary substantially in space and time. Such factors would need to be isolated to conclude what a "stable" community of borers or rate of bioerosion looks like.

Response: We consider "a stable bioeroding community" to reflect conditions when all ambient bioeroders – generally said microborers, grazers, macroborers – are present on/in the experimental substrates and the bioeroder assemblage does not significantly change anymore. As outlined by the reviewer, continuously fresh substrate becomes exposed and colonised, resulting in constant bioerosion rates. We added a short explanation in the text accordingly: "mature bioeroding community (i.e. a fully developed bioeroding community of microborers, macroborers, and grazers also present in the ambient environment, with relatively stable bioerosion rates) has established."

*9) Table 1:* Explain how the three replicates of the 7-year sample were made

Response: Was added accordingly: "*The 7-year block was cut into three replicates to obtain an impression of the spatial variability of bioerosion within a single block".

**Review 2 (Anonymous Referee)**

*1) Page 3 line 3.* 46 blocks of marble and limestone were used for the experiment and placed on the sea floor. You chose 20 blocks for your analysis. Could you detail the lithology of the chosen blocks?

Response: We noticed that both reviews requested more detail in the description of the experimental design. Please see our response to comment 1 of review 1.

*2) Page 3 line 13, page 8 line 17.* You cut blocks to a size of 10x10 cm. What is their thickness? What is the size of the blocks used for micro-CT? is it 90x90x18 mm as reported in the legends? How did you choose the parts of the blocks for micro-CT analysis?

Response: Please see again our response to comment 1 and 3 of review 1.

*3) Page 6 line 23- Cliona schmidti.* See comments for the legend of figure 3. line 24 and followings. When you quote the figure 6 with the spicules of the sponges, please quote also the corresponding figs of the bioerosion traces got with m-CT.

Response: A summary of the observed bioerosion traces and referred producers is provided in Table 2 and referenced at the beginning of chapter 3.2. In order to keep the information concise and clear, this comment was not implemented.

*4) Discussion page 8.* Could you consider if the lithology of the blocks affect the results of the erosion rates?

Response: Please see our response to comment 1 of review 1.

*5) Page 9 lines 15-18.* Please add comments about the dominance of sponges in the excavated blocks, discussing the succession phase of bioeroders. Sponges may precede worms and bivalves in colonisation (e.g. Hutchings, 1986). This could be true only for tropical waters, and moreover, these data about successional phases refer to shorter time gaps. But in temperate areas bioerosion rates are slow (for Teredinidae is demonstrated). Is it for this reason that worms are scarce? Moreover, these blocks are small (10x10 cm). Likely, they moved and rolled on the bottom. Worms (with singles openings) could have been smothered by mud, and the vacant holes occupied by sponges.

Response: Thank you very much for this valuable comment. We added information to the Discussion. The total absence of bioeroding bivalves and low occurrence of polychaetes during our experiment appears to be highly remarkable, but we assume that this is an effect of the high spatial variability of macrobioeroders. Because most of the deployed blocks were large enough to be stable under turbulent conditions, and smaller ones were anchored where necessary (please see comment 1 of review 1), movement of the blocks affecting the settlement of bioeroders is rather unlikely. We assume that the succession of bioeroder guilds in the Mediterranean Sea is very similar to that in the tropics with polychaetes being present after a few months, while sponges and molluscs may need several years to develop. However, the detection of organism succession would have required a more extended experimental design.

Content changes (page 10, line 3): Although on the surrounding sea floor (...) Other long-term experiments in the tropics indicated a distinct succession of macroborers with polychaetes being present after a few months, while sponges and molluscs needed several years to establish (Carreiro-Silva and McClanahan, 2012; Kiene and Hutchings, 1992, 1994; Osorno et al., 2005; Pari et al., 1998, 2002; Peyrot-Clausade et al., 1995). We assume that the dominance of sponges during the present study is (...)

*6) Page 10 line 20-25.* Is it your case? the different lithology of the blocks could have influenced also the erosion traces?

Response: Please see our response to comment 1 of review 1.

*7) Figure 1.* Show the Mediterranean sea and where Rhodes is.

Response: Figure changed accordingly.

*Figure 3.* Could be possible that in B two distinct traces are presents? Apart E. ovula, there are other cylindrical chambers arranged in chains, very similar to fig. 2C (E. cateriformis). In fact, the erosion traces in 2C are attributed to C. schmidti, and the traces in figure. 3B to two distinct species: Cliona viridis and C. schmidti. The chambers arranged in chains in Fig. 3C could belong to C. schmidti and the others, globose-ovoid to C. viridis. So the sentence " Cliona schmidtii (Ridley, 1881) was recognised as the trace maker of E. cateniformis and E. ovula " should be reconsidered.

Response: Thank you very much for this valuable comment. We agree that the cylindrical chambers crossing the main trace show a high resemblance to an early phase *Entobia cateniformis*. However, according to the original diagnosis in Bromley & D'Alessandro (1984), in its early phase *E. ovula* is similarly described to show "chambers that are separated from neighbours by a very short intercameral canal, usually reduced to a construction, which are arranged in straight strings".

According to this we consider the gallery to represent the earliest part of the *E. ovula* cluster, which was filled with the characteristic ovoid chambers in its later stage.

**Additional minor changes:**

Replaced "&" by "and" throughout.

Table 2, corrected missing value.

Figure 6: Abbreviation of *Cliona* after first account.

Supplement Table S1: Correction wrong assignment label Fig. 6 G and H in column B.

Supplement Table S2: Correction of automatic-correction errors in column "unassignable macroborings".

[revised manuscript text omitted]